# Remimazolam for anesthesia and sedation in pediatric ambulatory surgery: A scoping review protocol

Yi Zhang[ID][1⊙], Linyun Wang[2⊙], Shuang Guo[1], Qingjun Zeng[1], Haishan Cui[1], Yunbo Mo[ID][3*]

**1** Department of Anesthesiology, Maternal and Child Health Hospital of Wanzhou District, Chongqing City, People's Republic of China, **2** Nursing Department, Maternal and Child Health Hospital of Wanzhou District, Chongqing City, People's Republic of China, **3** Department of Pediatrics, Maternal and Child Health Hospital of Wanzhou District, Chongqing City, People's Republic of China

⊙ These authors contributed equally to this work

* moyunbo666@outlook.com

## Abstract

### Background

Remimazolam, an innovative benzodiazepine, exhibits potential for use in anesthesia and sedation for pediatric outpatient procedures, due to its rapid onset, predictable metabolism, and favorable safety profile. While adult studies are increasing, comprehensive evidence for pediatric use remains limited. This scoping review aims to systematically summarize and delineate the existing evidence concerning the application and features of remimazolam in anesthesia and sedation for outpatient pediatric surgical procedures.

### Objective

To systematically map existing evidence on remimazolam use in pediatric ambulatory surgery anesthesia and sedation, identifying key concepts, evidence sources, and knowledge gaps.

### Methods

Following JBI methodology and PRISMA-ScR guidelines, we will search multiple databases, as well as the recommendations provided in the Preferred Reporting Items for Systematic Reviews and Meta-Analyses extension tailored for Scoping Reviews (PRISMA-ScR) statement. A comprehensive search will be performed using multiple databases, which include PubMed/MEDLINE, Embase, the Cochrane Central Register of Controlled Trials (CENTRAL), Web of Science, CINAHL, Scopus, Google Scholar, along with Chinese databases like the China National Knowledge Infrastructure (CNKI), WanFang Data, and VIP Information.. All published studies on remimazolam use in patients ≤18 years for ambulatory surgery will be included. Two reviewers will independently screen and extract data using a standardized form.

**Data availability statement:** No datasets were generated or analysed during the current study. All relevant data from this study will be made available upon study completion.

**Funding:** The Joint General Fund of Science, Health Science and Technology of Wanzhou District, Chongqing, People's Republic of China, wzwjw-kw2024031, received by Mr. Yi Zhang.

**Competing interests:** The authors have declared that no competing interests exist.

## Expected results and dissemination

This review will provide a comprehensive evidence map of remimazolam in pediatric ambulatory anesthesia, highlighting research distribution, clinical applications, dosing strategies, and safety outcomes while identifying critical knowledge gaps for future research.

## Introduction

### Rationale

Remimazolam represents a promising advancement in pediatric anesthesia for ambulatory surgery. This ultra-short-acting benzodiazepine offers distinct advantages over traditional agents through its organ-independent metabolism by tissue esterases and rapid, predictable offset [1]. Clinical studies demonstrate superior procedural success rates and faster recovery compared to midazolam, with potentially better hemodynamic stability than propofol [2–6].

Although early studies have validated its safety and efficacy in clinical settings, there have been occasional case reports highlighting adverse events, including the possibility of re-sedation following flumazenil reversal [7,8]. Despite these advantages, pediatric evidence remains limited. Most research focuses on adults, with significant gaps in understanding remimazolam's use in specific pediatric populations, particularly infants and children with comorbidities.

Anesthesia in pediatric ambulatory surgery imposes rigorous requirements on anesthetic agents, necessitating a combination of quick onset, accurate controllability, rapid recovery, and a considerable safety margin. The pharmacokinetic characteristics of remimazolam, which is mainly metabolized by tissue esterases, suggest that it could be ideally suited to the needs of this unique pediatric demographic. Nevertheless, there is a lack of high-quality clinical research involving children, particularly infants under two years and those with significant comorbidities. This scarcity of evidence not only hinders the broader integration of remimazolam into pediatric clinical practice but also implies that clinical decisions associated with it lack a solid evidence foundation.

### Objectives

This scoping review aims to systematically map existing evidence on remimazolam use in pediatric ambulatory surgery. Our specific research questions are:

1. Which pediatric populations have been studied regarding remimazolam use?

2. In what clinical contexts and dosing strategies has pediatric remimazolam been evaluated?

3. What efficacy outcomes are reported in the literature?

4. What safety concerns have been identified?

5. How does remimazolam compare to standard agents?

6. What economic considerations are discussed?

7. What knowledge gaps require future investigation?

## Methods

This protocol follows JBI scoping review methodology [9] and PRISMA-P guidelines [10]. The final review will adhere to PRISMA-ScR reporting standards [11].

### Inclusion criteria

Using the PCC framework:

Population: Patients from birth to 18 years, all clinical conditions included.

Concept: Remimazolam use for anesthesia or procedural sedation in ambulatory surgery, including dosing protocols, efficacy measures, safety outcomes, and comparative effectiveness.

Context: All geographic regions and ambulatory surgical settings.

### Study types

This scoping review seeks to include a diverse array of published types of clinical research, such as, but not limited to: randomized controlled trials (RCTs), non-randomized controlled trials, cohort studies, case-control studies, cross-sectional studies, case series, and reports on individual cases. Additionally, systematic reviews and meta-analyses will be incorporated to identify their foundational studies. The inquiry will encompass relevant conference abstracts, clinical trial registry records, and reports of studies that have not been published. No restrictions will be imposed regarding the date of publication or the duration of the studies. Although literature in English and Chinese will be given preference, significant studies in other languages will be acknowledged, provided their content can be comprehensively understood using reliable translation tools (such as Google Translate or DeepL).

### Information sources

The subsequent electronic databases are set to be thoroughly examined:

PubMed/MEDLINE

Embase

Cochrane Central Register of Controlled Trials (CENTRAL)

Web of Science (Core Collection)

CINAHL (Cumulative Index to Nursing and Allied Health Literature)

Scopus

Google Scholar (first 200–300 results, as these typically contain the most relevant citations based on relevance ranking algorithms)

China National Knowledge Infrastructure (CNKI)

WanFang Data

VIP Chinese Science and Technology Journal Database (VIP)

Furthermore, manual searches will be conducted on the reference lists of the studies included, along with pertinent systematic reviews. Moreover, well-known clinical trial registries, including ClinicalTrials.gov, the EU Clinical Trials Register, and the Chinese Clinical Trial Registry (ChiCTR), will be examined to identify studies that are either currently in progress or have been completed but remain unpublished.

### Study status and timeline

This study protocol outlines a scoping review that has not yet commenced data collection or analysis. The planned timeline is as follows:

A) Literature search and screening (equivalent to participant recruitment for a primary study) is expected to be completed by June 30, 2025.

B) Data extraction and charting are expected to be completed by September 30, 2025.

C) Analysis of charted data and drafting of the final manuscript are expected to be completed by December 31, 2025.

All stages of this scoping review are currently in the planning phase, and no results have been generated or analyzed.

## Search strategy

Clinical trial registries will also be searched. The search strategy (Appendix A) combines controlled vocabulary and keywords.

## Study selection

After deduplication using EndNote, two reviewers will independently screen titles/abstracts and full texts using Rayyan for collaborative online screening. Disagreements will be resolved through discussion or third reviewer consultation.

## Data extraction

Two reviewers will independently extract data using a piloted standardized form (Table 1). The extracted information will undergo a cross-check, and any inconsistencies will be addressed through discussion, with a third reviewer stepping in for arbitration if required, to maintain accuracy and consistency. The extraction form will be initially tested on a small sample of representative articles and adjusted as necessary. The data that will be collected includes:

Examine overarching details of the research (including authors, publication year, country of origin, nature of the study, and its publication status).

Participant profile specifics (demographics, number of participants, age range, gender distribution, ASA classification, pre-existing conditions, and type of surgical procedure).

Details regarding the remimazolam intervention (indication for use, method of administration, dosage schedule, duration of treatment, and any additional medications).

Information about the control group intervention (if relevant).

Primary outcomes as reported (including effectiveness, safety considerations, pharmacokinetic/pharmacodynamic parameters, experiences of patients/operators, economic implications, and more).

Overall conclusions drawn by the study along with limitations recognized by the authors.

## Data synthesis and presentation of results

This scoping review is not intended to include a thorough assessment of bias risk in the selected studies or to carry out meta-analyses for synthesizing effect estimates. Rather, the data will be summarized in a narrative format, enhanced by tables and figures to effectively convey the distribution, attributes, fundamental concepts, and results of the studies.

The results will be organized based on the research questions formulated for this review, for instance:

Summarizing the overall features of the examined studies, which encompasses the quantity of studies, their categories, geographic spread, and patterns in the years of publication.

Reviewing how remimazolam is applied across various pediatric demographics and clinical situations.

Compiling the dosing regimens documented for remimazolam.

Charting the different outcome measures evaluated in the existing literature.

Highlighting specific knowledge gaps or potential avenues for future research within the current studies.

Quantitative data such as the count of studies, participant numbers, and age ranges will be assessed through descriptive statistical methods like frequencies and percentages.

**Table 1. Data extraction form – Remimazolam in pediatric day surgery anesthesia and sedation.**

| Data Category | Data Elements |
|---|---|
| **Study Characteristics** | Author(s)<br>Year of publication<br>Country of origin<br>Study design (e.g., RCT, observational study, case series/report)<br>Publication status (e.g., peer-reviewed journal, conference abstract)<br>Funding source |
| **Participant Characteristics** | Population type (e.g., healthy children, specific comorbidities)<br>Sample size (total and per group, if applicable)<br>Age (e.g., mean, SD, range, specific age subgroups like neonates, infants, toddlers, etc.)<br>Gender distribution<br>ASA physical status<br>Baseline comorbidities<br>Type of day surgery or procedure |
| **Intervention Details (Remimazolam)** | Purpose of remimazolam administration (e.g., procedural sedation, general anesthesia induction, general anesthesia maintenance)<br>Route of administration<br>Dosage regimen (e.g., bolus dose, infusion rate, total dose, titration protocol)<br>Duration of remimazolam administration<br>Concomitant medications (e.g., analgesics, other sedatives, neuromuscular blockers, reversal agents like flumazenil) |
| **Comparator Details (if applicable)** | Type of comparator (e.g., midazolam, propofol, dexmedetomidine, placebo)<br>Route of administration<br>Dosage regimen<br>Duration of administration<br>Concomitant medications in comparator group |
| **Outcome Measures & Main Findings** | **Efficacy Outcomes:**<br>Sedation/anesthesia success rate (as defined by study)<br>Onset time of sedation/anesthesia<br>Duration of action/ procedure time<br>**Recovery characteristics:**<br>Time to full alertness/ eye opening/ purposeful movement<br>Time to eligibility for Post-Anesthesia Care Unit (PACU) discharge<br>Time to hospital discharge<br>Quality of sedation/anesthesia (e.g., sedation scores, BIS values)<br>Incidence of emergence agitation/delirium<br>Need for rescue sedation/analgesia<br>Procedural conditions (e.g., patient movement, surgeon satisfaction)<br>**Safety Outcomes:**<br>Incidence and type of adverse events (e.g., respiratory depression, hypotension, bradycardia, nausea/vomiting, resedation)<br>Hemodynamic parameters (e.g., blood pressure, heart rate changes)<br>Respiratory parameters (e.g., oxygen saturation, respiratory rate, need for airway intervention)<br>Use of antagonists (e.g., flumazenil – dose, frequency, reason)<br>**Patient and Operator Experience:**<br>Patient satisfaction (if measured, e.g., parent/child report)<br>Operator/Anesthesiologist satisfaction (e.g., ease of use, satisfaction with conditions)<br>**Pharmacokinetic/Pharmacodynamic (PK/PD) Parameters:**<br>If reported (e.g., clearance, half-life, Cmax, Tmax, dose-response relationships)<br>**Economic Costs:**<br>If reported (e.g., drug cost, cost-effectiveness, length of stay impact) |
| **Key Conclusions & Limitations** | Main conclusions as stated by the authors<br>Author-identified limitations of the study<br>Identified knowledge gaps relevant to the scoping review questions |

RCT: Randomized Controlled Trial SD: Standard Deviation ASA: American Society of Anesthesiologists.

PACU: Post-Anesthesia Care Unit BIS: Bispectral Index PK/PD: Pharmacokinetic/Pharmacodynamic.

Cmax: Maximum Concentration Tmax: Time to Maximum Concentration.

## Discussion

This scoping review will provide the first comprehensive evidence map of remimazolam use in pediatric ambulatory surgery. Given the increasing demand for safe and efficient pediatric ambulatory anesthesia, understanding the current evidence landscape for novel agents like remimazolam is crucial. By systematically mapping available evidence, we aim to bridge the gap between emerging pharmacological innovations and clinical practice needs in pediatric settings.

### Anticipated findings across pediatric age groups

We anticipate finding extremely limited evidence for remimazolam use in infants under 2 years. This age group presents unique pharmacological challenges due to immature hepatic enzyme systems, altered volume of distribution, and heightened sensitivity to sedatives. The absence of robust data in this population likely reflects both ethical considerations in conducting research and regulatory caution. This gap is particularly concerning given that infants frequently require procedural sedation for imaging studies and minor procedures.

For preschool-aged children (2–5 years), we expect to identify a modest number of studies, primarily focusing on procedural sedation for non-invasive procedures. This age group's behavioral challenges and separation anxiety make effective sedation crucial. However, the evidence will likely be limited to small case series or single-center studies, highlighting the need for larger, multicenter investigations.

The majority of pediatric remimazolam studies will likely involve school-age children and adolescents (6–18 years), as this population more closely resembles adult physiology. We expect to find comparative studies with midazolam and possibly propofol in this age group, particularly for endoscopic procedures and minor surgeries. These studies may provide more robust efficacy and safety data, though generalizability to younger children will remain limited.

### Clinical practice implications

The clinical implications of our findings will extend across multiple domains of pediatric ambulatory practice. Regarding safety profile considerations, our review will clarify the safety margins of remimazolam across different pediatric populations, particularly regarding respiratory depression risk, hemodynamic stability, and recovery profiles. This information is essential for developing age-specific protocols and safety guidelines.

From an operational efficiency perspective, by mapping recovery times and discharge readiness data, our review will help ambulatory centers assess whether remimazolam can improve patient throughput without compromising safety. The rapid offset and predictable recovery profile suggested in adult studies could translate to significant operational advantages in high-volume pediatric settings.

Although formal economic analyses may be limited, our synthesis of dosing requirements, recovery times, and adverse event profiles will provide foundational data for future cost-effectiveness studies comparing remimazolam to current standard agents. This economic dimension is increasingly important as healthcare systems seek to optimize resource utilization while maintaining quality care.

Clinicians can use this review to identify patient populations with existing safety data, compare dosing strategies across different clinical contexts, understand reported adverse events for informed consent discussions, and recognize scenarios requiring additional caution due to limited evidence. These practical applications will support evidence-based decision-making in pediatric ambulatory settings.

### Factors affecting study outcomes

Several factors may influence the heterogeneity of findings in our review. Study design variations between randomized controlled trials and observational studies may yield different safety and efficacy profiles, with observational data potentially capturing real-world effectiveness but lacking the internal validity of controlled trials. Population differences

including age distribution, baseline health status, and procedure types will significantly affect drug response and outcome measures.

Geographic variations in practice patterns, regulatory approvals, and healthcare delivery systems may contribute to heterogeneity in dosing strategies and safety protocols. Additionally, the lack of standardized outcome measures across studies will complicate direct comparisons. Some studies may focus on clinical endpoints like sedation success rates, while others emphasize pharmacokinetic parameters or patient satisfaction metrics.

### Research priorities

Our systematic mapping will identify critical research priorities for future investigation. Dose-finding studies represent a fundamental need, as the absence of established pediatric dosing protocols across age groups remains a significant barrier to clinical adoption. Head-to-head comparative effectiveness trials with established agents (midazolam, propofol, dexmedetomidine) in specific procedural contexts are urgently needed to position remimazolam within existing treatment algorithms.

Special populations deserve focused investigation, including children with developmental disabilities, chronic conditions, or those requiring frequent procedures. These vulnerable groups often present unique sedation challenges and may respond differently to standard protocols. The potential neurodevelopmental impact of remimazolam, particularly with repeated exposures, requires longitudinal study designs to ensure long-term safety. Understanding genetic variations affecting remimazolam metabolism in diverse pediatric populations could optimize personalized dosing strategies and improve safety profiles.

### Study limitations

This scoping review has several inherent methodological limitations that readers should consider. Unlike systematic reviews, our scoping approach does not include formal quality assessment or risk of bias evaluation. This limitation prevents us from making definitive recommendations about intervention effectiveness but aligns with our objective of mapping the evidence landscape rather than synthesizing effect estimates.

The anticipated heterogeneity in study designs, populations, and outcome measures will preclude quantitative synthesis. While this diversity provides a comprehensive view of remimazolam applications, it limits our ability to draw firm conclusions about optimal use parameters. Given the recent introduction of remimazolam, our review captures an emerging evidence base that will likely expand rapidly. Regular updates to this review will be necessary to maintain clinical relevance.

While we include major English and Chinese databases, potentially relevant studies in other languages may be missed. This is particularly relevant given remimazolam's approval and use in Japan, where significant pediatric experience may exist in Japanese-language publications. To minimize this bias, we will use professional translation services for potentially relevant non-English/Chinese studies identified through reference screening, contact international colleagues to identify important regional publications, clearly acknowledge language limitations in our final review, and encourage future reviewers to expand language inclusion as resources permit.

### Novel contributions

This protocol addresses a critical evidence gap by providing the first systematic mapping of remimazolam use in pediatric ambulatory surgery. It will establish a foundation for future systematic reviews and guide research priorities in this emerging field. Through this comprehensive scoping review, we aim to provide a foundational evidence map that will guide both clinical decision-making and future research priorities in pediatric ambulatory anesthesia, ultimately contributing to safer and more effective sedation practices for children.

## Conclusion

This protocol outlines a strategy for conducting a scoping review intended to systematically map and describe the available evidence regarding the clinical application of remimazolam in anesthesia and sedation for pediatric outpatient surgical procedures. By engaging in this process, we aim to enhance our understanding of the present research environment, pinpoint critical knowledge deficiencies, and offer significant insights for advancing clinical practices and future scientific exploration.

## Supporting information

**S1 Appendix.  PubMed search strategy.**
(DOCX)

**S1 Checklist.  PRISMA-P 2015 checklist.**
(DOCX)

**S1 Fig.  PRISMA flow diagram.**
(PDF)

**S2 Checklist.  PRISMA 2020 scoping review.**
(DOCX)

## Acknowledgments

The work presented in this paper was greatly enhanced by the support of numerous individuals. We sincerely thank all the members of the research team for their thoughtful guidance and valuable suggestions during the processes of selecting the topic, developing the framework, and making revisions. Our appreciation also goes to the Wanzhou District Health Commission and the Science and Technology Bureau of Chongqing City for their collaborative financial support. Lastly, we are thankful to our families for their steadfast understanding and companionship.

## Author contributions

**Conceptualization:** Yi Zhang, Yunbo Mo.

**Investigation:** Qingjun Zeng, Haishan Cui, Shuang Guo.

**Methodology:** Linyun Wang, Yunbo Mo.

**Supervision:** Yunbo Mo.

**Validation:** Haishan Cui, Shuang Guo.

**Visualization:** Qingjun Zeng.

**Writing – original draft:** Yi Zhang, Linyun Wang.

**Writing – review & editing:** Yi Zhang, Linyun Wang, Qingjun Zeng, Haishan Cui, Shuang Guo, Yunbo Mo.

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
