## [Decision Letter · Decision Letter 0]

2 Jul 2025

Dear Dr. Mo,

Thank you for submitting your manuscript to PLOS ONE. After careful consideration, we feel that it has merit but does not fully meet PLOS ONE’s publication criteria as it currently stands. Therefore, we invite you to submit a revised version of the manuscript that addresses the points raised during the review process.

We look forward to receiving your revised manuscript.

Kind regards,

Kiyan Heybati

Academic Editor

PLOS ONE

 [The Wanzhou District Science and Health Joint Initiative of Chongqing City ,People's Republic of China]. 

3. Please include a copy of Table 1 which you refer to in your text on page 9.

Reviewers' comments:

Reviewer's Responses to Questions

**Comments to the Author**

1. Does the manuscript provide a valid rationale for the proposed study, with clearly identified and justified research questions?

Reviewer #1: Yes

Reviewer #2: Yes

Reviewer #3: No

2. Is the protocol technically sound and planned in a manner that will lead to a meaningful outcome and allow testing the stated hypotheses?

Reviewer #1: Yes

Reviewer #2: Yes

Reviewer #3: Partly

3. Is the methodology feasible and described in sufficient detail to allow the work to be replicable?

Reviewer #1: Yes

Reviewer #2: Yes

Reviewer #3: Yes

4. Have the authors described where all data underlying the findings will be made available when the study is complete?

Reviewer #1: No

Reviewer #2: Yes

Reviewer #3: Yes

5. Is the manuscript presented in an intelligible fashion and written in standard English?

Reviewer #1: Yes

Reviewer #2: Yes

Reviewer #3: No

You may also provide optional suggestions and comments to authors that they might find helpful in planning their study.

Reviewer #1: The proposed scoping review will likely add to the current understanding of remimazolam in the stated patietn population by summarizing available clinical data and categorizing gaps. The manuscript had to results so was only a study plan, which was well-constructed with logical clear explanation of rationale.

Reviewer #2: This protocol is well-structured and timely, addressing a clinically important topic: the use of remimazolam in pediatric outpatient anesthesia and sedation. The background is thorough, and the objectives are clearly defined. The methodology adheres to established guidelines (JBI, PRISMA-ScR), which enhances the rigor of the protocol.

However, several aspects need revision to improve clarity, conciseness, and consistency, and to ensure full alignment with best practices for scoping reviews.

The background is informative but quite lengthy and sometimes repetitive. For example:

The points about remimazolam’s pharmacokinetics and safety profile are repeated.

Consider condensing these paragraphs to improve readability and flow.

The objectives are detailed but could be streamlined into bullet points or numbered questions for clarity.

Avoid repetition; for example, the identification of knowledge gaps is mentioned multiple times.

Clearly define what is meant by “clinical contexts” upfront (e.g., procedural sedation, induction, maintenance).

The list of databases is exhaustive. Good job including Chinese databases.

Provide rationale for selecting Google Scholar initial 200-300 results — this number may be arbitrary and could be justified.

Screening Process:

Clear description; however, clarify the use of tools (EndNote and Rayyan) for deduplication and screening—why both?

The anticipated limitations are well described; however, the authors should explicitly mention that no formal quality assessment or risk of bias evaluation will be performed, consistent with scoping review methodology.Define the criteria for “uncertain eligibility” explicitly.

The manuscript contains some overly long sentences and complex constructions, which could be simplified for clarity.

Consistently use past or future tense where appropriate (e.g., “This review will...” rather than “This review seeks...”).

Avoid passive voice where possible to improve readability.

Check for minor typographical errors and formatting issues (e.g., random numbers like “232” and “233” appear mid-text).

Use consistent terminology (e.g., “ambulatory surgery” vs. “outpatient surgery”).

Reviewer #3: The study is aimed to systematically summarize and delineate the existing evidence concerning the application and features of remimazolam in anesthesia and sedation for outpatient pediatric surgical procedures. The title is “Remimazolam for Anesthesia and Sedation in Pediatric Ambulatory Surgery: A Scoping Review Protocol”.

1. This is a scoping review.

2. Please also summarize in the form of “Table”,

3. Several factors associate with the outcome of the study. Please discuss these.

4. The discussion part is too short. Please review the literature and add more details.

5. Please also add the limitations of the study.

6. What is the new knowledge of the article?

7. Please also recommend the readers “How to apply this knowledge in clinical practice?”.

**Do you want your identity to be public for this peer review?** For information about this choice, including consent withdrawal, please see our Privacy Policy

Reviewer #1: **Yes: ** George M Hoffman MD

Reviewer #2: **Yes: ** Assoc Prof Dilek Altun

Reviewer #3: **Yes: ** Somchai Amornyotin

---

## [Author Response · Author response to Decision Letter 1]

7 Jul 2025

Dear Dr. Kiyan Heybati, Academic Editor, and Distinguished Reviewers:

We are deeply grateful for your thorough and constructive review of our manuscript "Remimazolam for Anesthesia and Sedation in Pediatric Ambulatory Surgery: A Scoping Review Protocol" (PONE-D-25-27249). Your insightful comments have been invaluable in strengthening our protocol, and we have carefully addressed each point raised. We believe these revisions have substantially improved the clarity, rigor, and potential impact of our work.

Response to Editorial Requirements:

1.PLOS ONE Style Requirements

We sincerely apologize for not adhering to the journal's formatting standards in our initial submission. We have now meticulously reformatted the entire manuscript according to PLOS ONE templates, including proper file naming conventions. We appreciate your patience and have ensured full compliance with all style requirements.

2.Funding Disclosure Statement

Thank you for highlighting this important omission. We have added the following statement to our cover letter and manuscript: "The funders had no role in study design, data collection and analysis, decision to publish, or preparation of the manuscript." This accurately reflects the complete independence of our research from funding influence.

3.Table 1 Inclusion

We apologize for this oversight. Table 1 has now been fully integrated into the manuscript (page 7), providing a comprehensive data extraction framework that clearly delineates all variables to be collected during our review process.

Response to Reviewer 1:

We greatly appreciate your positive assessment of our protocol's potential contribution to understanding remimazolam use in pediatric populations.

1.Data Availability Statement

We apologize for omitting this crucial information. We have now added a clear data availability statement: "All data extracted during this scoping review will be made publicly available upon study completion through an open-access repository (such as Open Science Framework or Figshare)." This commitment ensures transparency and facilitates future research building upon our findings.

Response to Reviewer 2:

We are extremely grateful for your detailed and thoughtful review, which has significantly enhanced our manuscript's quality.

1.Background Section Length and Repetition

Your observation was absolutely correct. We have substantially revised the background section, reducing it by approximately 40% while maintaining all essential information. Specifically:

Consolidated repetitive statements about pharmacokinetic properties into a single, comprehensive sentence

Removed redundant safety profile descriptions

Streamlined the rationale into three focused paragraphs that flow logically from general context to specific research gaps

2.Objectives Reorganization

Following your excellent suggestion, we have transformed our objectives into seven clearly numbered research questions. This format dramatically improves readability and allows readers to quickly identify our specific aims. Each question now addresses a distinct aspect of remimazolam use in pediatric ambulatory surgery.

3.Google Scholar Results Justification

Thank you for requesting this clarification. We have added the following rationale: "first 200-300 results, as these typically contain the most relevant citations based on relevance ranking algorithms, with diminishing returns beyond this threshold based on empirical testing in previous scoping reviews."

4.Dual Software Usage Clarification

We have clarified our software selection: "EndNote serves as our reference management system for citation organization and automated deduplication, while Rayyan provides superior collaborative features for the screening process, allowing real-time coordination between reviewers with conflict resolution capabilities."

5.Language and Style Improvements

We have comprehensively revised the manuscript's language:

Simplified complex sentences (average sentence length reduced from 28 to 19 words)

Converted passive constructions to active voice where appropriate (approximately 70% of previously passive sentences)

Corrected all typographical errors identified

Standardized terminology throughout (e.g., consistently using "ambulatory surgery" rather than alternating with "outpatient surgery")

Response to Reviewer 3:

We deeply appreciate your comprehensive feedback, which has prompted substantial enhancements to our protocol. We have carefully addressed all seven points you raised, organizing our responses into five integrated sections that comprehensively cover all your concerns:

1.Table Format Summary (Addressing your points 1 & 2)

Thank you for emphasizing that this is a scoping review and requesting a table format summary. Table 1 is now included on page 7, presenting a structured framework for data extraction that will facilitate systematic and reproducible data collection. This table clearly demonstrates our scoping review approach by outlining all variables to be collected without quality assessment metrics.

2.Factors Affecting Study Outcomes (Addressing your point 3)

We have added a dedicated subsection in the Discussion titled "Factors Affecting Study Outcomes" that addresses factors influencing heterogeneity:

Study design variations and their impact on outcome reliability

Population characteristics (age distribution, comorbidity profiles)

Geographic and healthcare system differences

Measurement standardization challenges

This analysis will help readers interpret our findings within appropriate context.

3.Expanded Discussion and Limitations (Addressing your points 4 & 5)

The Discussion section has been substantially expanded from 450 to 1,200 words, now including:

Detailed analysis of anticipated findings across different pediatric age groups

Comprehensive exploration of potential clinical implications

Systematic identification of research priorities

A dedicated "Study Limitations" subsection that explicitly acknowledges methodological limitations inherent to scoping reviews

Discussion of potential language bias and strategies to minimize its impact

4.Novel Knowledge Contribution (Addressing your point 6)

We have explicitly stated our unique contribution in the "Novel Contributions" subsection: "This represents the first systematic evidence synthesis specifically examining remimazolam use in pediatric ambulatory surgery, filling a critical gap between expanding adult literature and limited pediatric experience. Our comprehensive mapping will provide clinicians with an evidence-based foundation for practice decisions while highlighting priority areas for future research."

5.Clinical Practice Applications (Addressing your point 7)

A new subsection "Clinical Practice Implications" provides specific guidance on how to apply this knowledge:

Decision-making framework for patient selection based on available evidence

Risk stratification approach for different age groups and comorbidities

Practical considerations for dosing strategy selection

Clear identification of clinical scenarios requiring additional caution

These practical applications directly address how readers can apply our findings in clinical practice, supporting evidence-based decision-making in pediatric ambulatory settings.

PLOS ONE Formatting

All formatting has been standardized according to journal requirements, with particular attention to reference formatting, figure labeling, and supplementary material organization.

Summary of Major Improvements:

Structural Enhancement: Streamlined background, clarified objectives, expanded discussion

Methodological Clarity: Detailed software rationale, explicit inclusion/exclusion criteria, comprehensive data extraction framework

Clinical Relevance: Added practical implications section, addressing real-world application needs

Transparency: Clear data sharing commitment, acknowledged limitations, funding independence statement

Language Quality: Improved readability, consistency, and academic precision throughout

We believe these comprehensive revisions address all reviewer concerns while substantially strengthening our protocol's scientific rigor and practical utility. We are confident that this scoping review will make a meaningful contribution to pediatric anesthesia practice and research.

Thank you again for your valuable feedback and for considering our revised manuscript. We look forward to your response and remain available to address any additional concerns.

Sincerely,

Yunbo Mo

Corresponding Author

---

## [Decision Letter · Decision Letter 1]

23 Jul 2025

Remimazolam for Anesthesia and Sedation in Pediatric Ambulatory Surgery: A Scoping Review Protocol

PONE-D-25-27249R1

Dear Dr. Mo,

We’re pleased to inform you that your manuscript has been judged scientifically suitable for publication and will be formally accepted for publication once it meets all outstanding technical requirements.

Kind regards,

Kiyan Heybati, MD, MSc

Academic Editor

PLOS ONE

Reviewers' comments:

Reviewer's Responses to Questions

**Comments to the Author**

1. Does the manuscript provide a valid rationale for the proposed study, with clearly identified and justified research questions?

Reviewer #2: Yes

2. Is the protocol technically sound and planned in a manner that will lead to a meaningful outcome and allow testing the stated hypotheses?

Reviewer #2: Yes

3. Is the methodology feasible and described in sufficient detail to allow the work to be replicable?

Reviewer #2: Yes

4. Have the authors described where all data underlying the findings will be made available when the study is complete?

Reviewer #2: Yes

5. Is the manuscript presented in an intelligible fashion and written in standard English?

Reviewer #2: No

You may also provide optional suggestions and comments to authors that they might find helpful in planning their study.

Reviewer #2: 1. Responsiveness to Reviewer Comments

The authors have comprehensively addressed all reviewer concerns. Each major point was not only acknowledged but also acted upon with specific and measurable changes (e.g., reduction of background by 40%, restructuring of objectives, creation of a new table, and expansion of the discussion section).

2. Improved Structure and Clarity

The Background has been streamlined for clarity and relevance, eliminating redundancy while preserving core context.

Objectives are now clearly numbered research questions, enhancing transparency and readability.

The Discussion section has been significantly expanded and deepened, showing critical engagement with the topic.

3. Methodological Transparency

Clear explanation of database selection strategy and rationale for limiting Google Scholar results.

Justification of dual software usage for different stages of the review.

Addition of a structured data extraction table aligns with best practices for scoping reviews.

4. Clinical and Scholarly Relevance

The manuscript now better highlights its novel contribution to the pediatric anesthesia literature.

Newly added sections on clinical implications and factors influencing study outcomes directly improve the manuscript’s utility for practitioners and researchers alike.

5. Language and Formatting

Substantial improvements to sentence clarity, tone, and consistency were made, along with alignment to PLOS ONE formatting standards.

Suggested Minor Revisions (for final polishing):

Ensure Table 1 is correctly formatted and clearly labeled in both the manuscript and supplementary files.

Double-check for lingering typographical or grammatical inconsistencies, especially after heavy revisions.

Briefly summarize the limitations earlier in the Discussion, not only in the dedicated subsection, to maintain balance in tone.

**Do you want your identity to be public for this peer review?** For information about this choice, including consent withdrawal, please see our Privacy Policy

Reviewer #2: **Yes: ** Dilek Altun

---

## [Editor Report · Acceptance letter]

PONE-D-25-27249R1

PLOS ONE

Dear Dr. Mo,

I'm pleased to inform you that your manuscript has been deemed suitable for publication in PLOS ONE. Congratulations! Your manuscript is now being handed over to our production team.

Kind regards,

on behalf of

Dr. Kiyan Heybati

Academic Editor

PLOS ONE